# EgoDemoGen: Novel Egocentric Demonstration Generation Enables Viewpoint-Robust Manipulation

## Abstract

Imitation learning based policies perform well in robotic manipulation, but they often degrade under *egocentric viewpoint shifts* when trained from a single egocentric viewpoint. To address this issue, we present **EgoDemoGen**, a framework that generates *paired* novel egocentric demonstrations by retargeting actions in the novel egocentric frame and synthesizing the corresponding egocentric observation videos with proposed generative video repair model **EgoViewTransfer**, which is conditioned by a novel-viewpoint reprojected scene video and a robot-only video rendered from the retargeted joint actions. EgoViewTransfer is finetuned from a pretrained video generation model using self-supervised double reprojection strategy. We evaluate EgoDemoGen on both simulation (RoboTwin2.0) and real-world robot. After training with a mixture of EgoDemoGen-generated novel egocentric demonstrations and original standard egocentric demonstrations, policy success rate improves **absolutely** by **+17.0%** for standard egocentric viewpoint and by **+17.7%** for novel egocentric viewpoints in simulation. On real-world robot, the **absolute** improvements are **+18.3%** and **+25.8%**. Moreover, performance continues to improve as the proportion of EgoDemoGen-generated demonstrations increases, with diminishing returns. These results demonstrate that EgoDemoGen provides a practical route to egocentric viewpoint-robust robotic manipulation.

## 1 Introduction

Imitation learning has emerged as a powerful paradigm in robotic manipulation, enabling end-to-end visuomotor policies that map raw observations to control actions. Recent imitation learning policies including Vision-Language-Action models (Chi et al., 2023; Zhao et al., 2023; Zitkovich et al., 2023; Ghosh et al., 2024; Liu et al., 2024; O'Neill et al., 2024; Black et al., 2024) have demonstrated remarkable performance when trained on large and diverse demonstration datasets (Wu et al., 2024; Khazatsky et al., 2024; Walke et al., 2023; O'Neill et al., 2024). However, these policies remain sensitive to distribution shift: policies trained or finetuned from a single egocentric viewpoint often fail to generalize to unseen egocentric viewpoints (Tian et al., 2025; Xing et al., 2025), See Figure 1(a). This limitation underscores the need to increase viewpoint diversity.

Generating *novel egocentric viewpoint* demonstrations serves as one effective solution to this problem. Existing efforts to mitigate this issue can be broadly categorized into two lines of work. One line of works focus on novel viewpoint synthesis using techniques such as point cloud rendering, 3D reconstruction, or image generation models (Sargent et al., 2024; Xue et al., 2025; Yang et al., 2025). These approaches synthesize novel visual observations but maintain original actions, leading to visual-action mismatch in egocentric setting, shown in Figure 1(b). Another line of works employ world models or action-conditioned video generation to target prediction or planning, rather than observation-action paired demonstration generation (Wang et al., 2025a; Rigter et al., 2024; Bruce et al., 2024; Luo & Du, 2024; Hafner et al., 2025). Moreover, these works do not explicitly model changes in the egocentric viewpoint caused by robot motion. Generating demonstrations from a novel egocentric viewpoint requires coherent synthesis of both the visual observations and the corresponding actions.

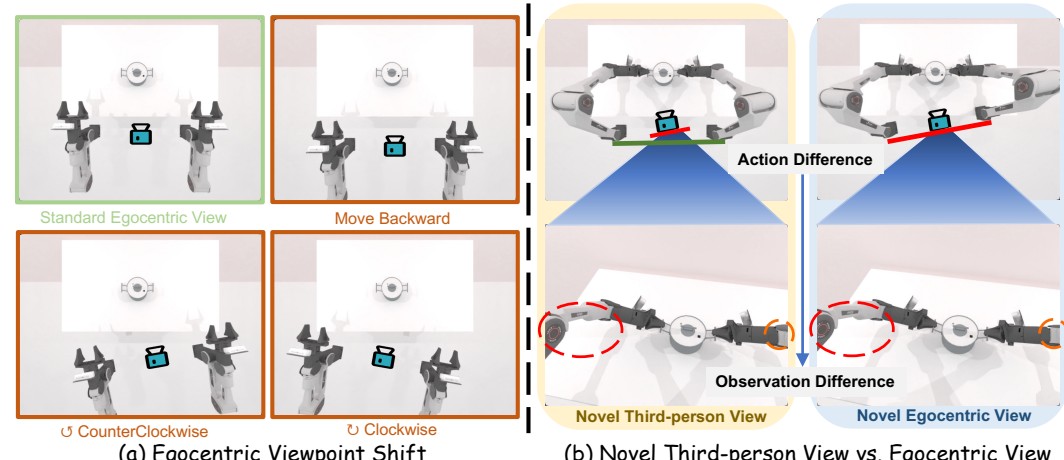

Standard Egocentric View     Move Backward

↺ CounterClockwise     ↻ Clockwise

(a) Egocentric Viewpoint Shift

Action Difference

Observation Difference

Novel Third-person View     Novel Egocentric View

(b) Novel Third-person View vs. Egocentric View

Figure 1: **Illustration of viewpoint transformations.** (a) Shifts in the egocentric viewpoint, including backward translation and clockwise/counterclockwise rotations. (b) Compared with a third-person view, robot base link and egocentric camera are mechanically coupled under egocentric view. A novel egocentric view requires action retargeting and observation synthesis consistent with the retargeted robotic arm state.

Our key insight addresses this fundamental gap: generating novel egocentric demonstrations requires not only synthesizing realistic observations from novel egocentric viewpoints, but also retargeting the original actions to align with the shifted viewpoint. This entails tackling two core challenges: (1) producing kinematically feasible robot actions that achieve the task under the novel egocentric viewpoint, and (2) generating realistic, temporally consistent observation videos that match these retargeted actions. Crucially, the generated demonstrations must preserve the style and intent of the original demonstrations while ensuring visual–action alignment.

To tackle these challenges, we propose **EgoDemoGen**, a novel framework for generating demonstrations from novel egocentric viewpoints. First, on the action side, we perform kinematics-based action retargeting to produce joint actions corresponding to novel egocentric viewpoint. Second, on the visual side, we propose **EgoViewTransfer**, a generative vide repair model that fuses the reprojected scene videos with the robot motion videos rendered under the retargeted actions and generate novel egocentric observation videos. We conducted experiments in the RoboTwin2.0 (Chen et al., 2025b) simulation environment and on a real-world dual-arm robot to evaluate the effectiveness of the generated demonstrations.

Our main contributions can be summarized as follows:

- We present **EgoDemoGen**, a framework that generates novel egocentric demonstrations with *paired* retargeted actions and egocentric observation videos, improving policy generalization to egocentric viewpoint shifts.

- We generate novel demonstrations by retargeting actions in the *novel* egocentric frame and synthesizing corresponding observation videos from a novel-viewpoint *reprojected scene video* and a *robot-only video* rendered from the retargeted joint actions. The generated *paired* demonstrations are used to train downstream policies.

- We introduce **EgoViewTransfer**, a *generative video repair model* finetuned from a pretrained video generation model with a *double reprojection* strategy, which fuses reprojected scene video and rendered robot video to synthesize consistent, realistic egocentric observation video.

- Experiments on simulation (RoboTwin2.0) and real-world robot show policy success rate **absolute** improvements of **+17.0%** (standard egocentric viewpoint) and **+17.7%** (novel egocentric viewpoints) in simulation when incorporating demonstrations generated by EgoDemoGen into the training mixture, and **+18.3%** and **+25.8%** on real-world robot.

Moreover, performance continues to improve as the proportion of **EgoDemoGen**-generated demonstrations increases, with diminishing returns.

## 2 RELATED WORK

**Data Generation for Policy Learning.** Generating synthetic demonstrations has emerged as a promising approach to address data scarcity in imitation learning (Hussein et al., 2017; Zhao et al., 2020). Existing methods can be broadly categorized into three streams: (1) **Geometry-based methods** leverage 3D representations to generate novel viewpoint observations but typically preserve original actions, causing visual-action mismatch under significant viewpoint changes (Xue et al., 2025; Yang et al., 2025; Zhou et al., 2023). (2) **Visual synthesis methods** focus on generating realistic observations without corresponding actions via advanced generative image models, making them unsuitable for egocentric demonstration generation (Tian et al., 2025; Sargent et al., 2024; Chen et al., 2023; 2025a). (3) **Motion retargeting methods** excel at adapting action trajectories to new object poses but operate primarily from third-person viewpoint without addressing egocentric viewpoint shifts (Mandlekar et al., 2023; Ameperosa et al., 2025; Lin et al., 2025). Unlike these works, EgoDemoGen specifically addresses egocentric demonstration generation by utilizing a generative video repair model EgoViewTransfer that synthesizes novel viewpoint observations conditioned on both reprojected scene videos and robot motion videos, while simultaneously retargeting actions to maintain precise visual-action alignment from the novel egocentric viewpoint.

**Video Generation Models in Robotics.** Recent advances in foundational video generation models have catalyzed applications in robotics (Blattmann et al., 2023; Hong et al., 2022; Yang et al., 2024; Zhu et al., 2024b). Broadly, methods fall into three groups: (1) **video-as-policy** train or adapt a strong video model and decode executable actions from its rollouts, yielding policies with improved generalization (Cheang et al., 2024; Liang et al., 2025b;a). (2) **predict-then-act** first synthesize future observations (often text/goal-conditioned) and then infer actions or supervision from the generated sequences (Du et al., 2023; Luo & Du, 2024; Patel et al., 2025). (3) **action-conditioned world models** generate future video conditioned on actions to serve as simulators or data engines (Zhu et al., 2024a; Zhou et al., 2024; Liu et al., 2025; Jang et al., 2025). Unlike these directions, which rarely handle novel egocentric viewpoint shifts explicitly, EgoDemoGen finetunes a video generation model to produce observation videos paired with retargeted actions in the novel egocentric frame, ensuring visual–action alignment for egocentric viewpoint-robust learning.

## 3 METHODOLOGY

### 3.1 OVERVIEW

Our goal is to generate novel egocentric demonstration pairs $(\tilde{V}, \tilde{Q})$ for a dataset of demonstrations $\{(V, Q, D)\}$ collected from single egocentric viewpoint. Each demonstration includes an RGB observation video $V$ and the corresponding robot joint actions $Q$, along with aligned depth maps $D$. A novel egocentric viewpoint $v$ is defined by a movement $(\Delta x, \Delta y, \Delta\theta)$ of the robot base, where $\Delta x, \Delta y$ are translations and $\Delta\theta$ is a rotation about its vertical axis. For any such novel egocentric viewpoint, our framework **EgoDemoGen** generates a paired demonstration comprising a egocentric observation video $\tilde{V}$ and its corresponding kinematically feasible joint actions $\tilde{Q}$. The detailed **EgoDemoGen** framework is shown in Figure 2

The proposed **EgoDemoGen** operates through two main modules: (1) **Action Retargeting** (§3.2): retarget actions in the *novel* egocentric frame. (2) **Novel Egocentric Observation Generation** (§3.3): synthesize novel egocentric observation with **EgoViewTransfer** by constructing a novel-viewpoint reprojected *scene video* and a *robot-only video* rendered from the retargeted joint actions as inputs. **EgoViewTransfer** (§3.4) is finetuned from a pretrained video generation model using a self-supervised *double reprojection* strategy.

### 3.2 ACTION RETARGETING

The objective of action retargeting is to compute a kinematically feasible joint actions $\tilde{Q}$ that reproduces the original task from the novel egocentric viewpoint $v$.

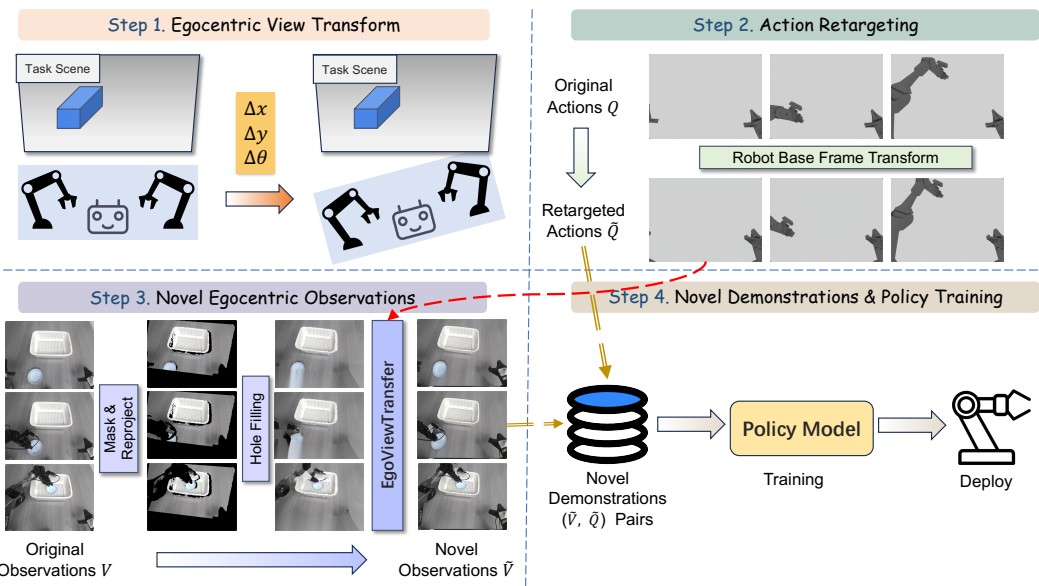

Figure 2: **Overview of EgoDemoGen. (1) Egocentric View Transform:** a *Novel Egocentric View* is specified by robot base motion $(\Delta x, \ \Delta y, \ \Delta \theta)$. **(2) Action Retargeting:** the original joint actions $Q$ is *retargeted* into the novel robot base frame to yield a kinematically feasible joint actions $\tilde{Q}$. **(3) Novel Egocentric Observations:** starting from the original observation video $V$, we mask the robot, reproject the scene to the novel viewpoint, perform hole filling, and apply **EgoViewTransfer** to synthesize the coherent observations $\tilde{V}$. **(4) Novel Demonstrations & Policy Training:** we obtain aligned pairs $(\tilde{V}, \ \tilde{Q})$ for training egocentric viewpoint-robust policies.

For each arm $a \in \{L, R\}$, time step $t$ and $q_t^a \in Q$, we first compute the end-effector pose in the source base frame via forward kinematics:

$$T_e^a(t) = \text{FK}_a(q_t^a). \tag{1}$$

This pose is then transformed into the target base frame defined by $v$:

$$T_e^{a \to v}(t) = T_v^a \cdot T_e^a(t). \tag{2}$$

We subsequently solve inverse kinematics to find the corresponding joint angles in the target frame:

$$q_t^{\prime a} = \text{IK}_a(T_e^{a \to v}(t)). \tag{3}$$

The retargeted dual-arm sequence is finally obtained by concatenating the joint angles for both arms: $\tilde{Q} = \{[q_t^{\prime L}; q_t^{\prime R}]\}_{t=1}^T$. Further details can be found in Appendix A.2

### 3.3 NOVEL EGOCENTRIC OBSERVATION GENERATION

The generation of the novel egocentric observation video $\tilde{V}$ aligned with the retargeted joint actions $\tilde{Q}$ proceeds through three main stages. Given the source RGB-D sequence $(V, D)$ and the novel egocentric viewpoint $v$, the pipeline first prepares a scene video by reprojecting the scene and applying hole filling, then renders a robot video from the retargeted joint actions, and finally fuses both through conditional generative video repair to produce the final output.

**Scene Video Preparation.**     Since the robot's joint state differs in the novel egocentric viewpoint, We begin by removing the robot from the original frames to isolate the scene content. We render a segmentation mask $M_t^{\text{robot}}$ for each frame, details in Appendix A.3:

$$I_t^{\text{scene}} = (1 - M_t^{\text{robot}}) \odot I_t. \tag{4}$$

The scene-depth pair $(I_t^{\text{scene}}, D_t)$ is reprojected to novel egocentric viewpoint, details in Appendix A.4. This reprojection creates an initial novel-view scene image $\hat{I}_t^{\text{nov}}$ but introduces holes due to occlusion. We apply hole filling to complete the missing regions, producing $I_t^{\text{scene,nov}}$. The scene video $V_S^{\text{nov}}$ is formed by aggregating these frames.

**Robot Video Rendering.** Using the retargeted joint action sequence $\tilde{Q}$ and the novel viewpoint parameters, we render a robot-only video $V_R^{\text{nov}} = \{I_t^{\text{robot,nov}}\}_{t=1}^T$, details in Appendix A.3. This video serves as a conditioning input to the generative video repair model, providing information about the robot's motion in the novel egocentric viewpoint.

**Conditional Video Generative Repair.** The final novel-view video $\tilde{V}$ is generated by a conditional generative video repair model **EgoViewTransfer** (described in §3.4) $G_\phi$ that takes both the scene video $V_S^{\text{nov}}$ and robot video $V_R^{\text{nov}}$ as inputs:

$$\tilde{V} = G_\phi(V_S^{\text{nov}}, V_R^{\text{nov}}). \tag{5}$$

The generated video $\tilde{V}$ shows a complete execution of the task from the novel egocentric viewpoint, with the robot motion aligned to $\tilde{Q}$.

## 3.4 EGOVIEWTRANSFER

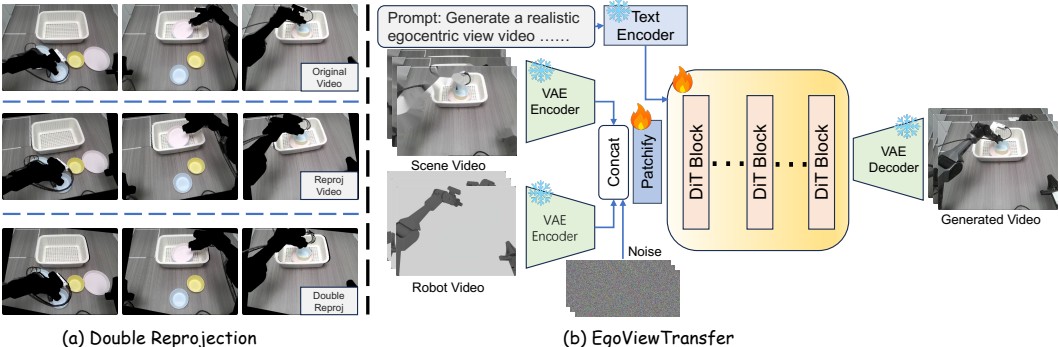

(a) Double Reprojection  (b) EgoViewTransfer

Figure 3: **EgoViewTransfer. (a) Double reprojection.** It simulates artifacts and occlusions caused by viewpoint change. The double reprojected video are aligned with the original video to form input/label pairs for training. **(b) Architecture of EgoViewTransfer.** The model takes a degraded scene video and a robot video as conditions and generates egocentric observation videos consistent with dual inputs.

We employ a self-supervised finetuning strategy to train EgoViewTransfer $G_\phi$ using only source egocentric view demonstrations, eliminating the need for novel egocentric view ground truth data. Our approach adapts the double reprojection technique (YU et al., 2025) to simulate the occlusions encountered during novel view synthesis.

**Double Reprojection for Data Preparation.** For each source RGB-D frame $(I_t, D_t)$, we simulate the geometric distortions and occlusion artifacts typical of novel view synthesis through a double reprojection process, see Figure 3(a). We first reproject the source frame to randomly sampled novel viewpoint $v \sim p(v)$ and obtain novel view frame $(\hat{I}_t^{\text{nov}}, \hat{D}_t^{\text{nov}})$. To simulate incomplete scene information, we repeoject this novel view frame back to he original source view and obtain artifacted scene frames $\check{V}_S^{\text{src}} = \{\check{I}_t^{\text{src}}\}_{t=1}^T$ that mimic the challenges of novel-view synthesis. Details of novel view reprojection can be found in Appendix A.4.

**Training Objective.** To mitigate the influence of robot-shaped black holes in the artifacted scene video $\check{V}_S^{\text{src}}$, which result from the original robot mask and may misalign with the retargeted robot

actions, we apply hole filling to obtain the final scene conditioning video $\bar{V}_S^{\text{src}}$. The model then learns to reconstruct the original source video $V^{\text{GT}} = \{I_t\}_{t=1}^T$ conditioned on both $\bar{V}_S^{\text{src}}$ and the source robot video $V_R^{\text{src}}$ (rendered from the original joint action sequence $Q$), see Figure 3(b). Following the standard diffusion training paradigm, we minimize the denoising objective:

$$\mathcal{L}_{\text{diff}} = \mathbb{E}_{\tau,\varepsilon}\|\varepsilon - \hat{\varepsilon}_\phi(\mathbf{x}_\tau, \tau; [\bar{V}_S^{\text{src}}, V_R^{\text{src}}])\|_2^2, \tag{6}$$

where $\mathbf{x}_\tau = \alpha_\tau V^{\text{GT}} + \sigma_\tau \varepsilon$ is a noisy version of the target video.

**Inference.** During inference at a novel egocentric viewpoint $v$, the generation pipeline is consistent with *Conditional Video Generative Repair* in §3.3.

## 4 EXPERIMENT RESULTS

### 4.1 EXPERIMENTAL SETUP

#### 4.1.1 ENVIRONMENTS AND TASKS

We conduct experiments in both the RoboTwin2.0 (Chen et al., 2025b) simulation environment and on a dual-arm robot in the real world, as illustrated in Figure 4.

**Simulation Environment:** We conduct simulation experiments in RoboTwin2.0 with dual Piper arms. Detailed Setup is in Appendix A.5.1. We evaluate on three tasks: *Lift Pot*, *Handover Mic*, and *Shake Bottle*. 100 demonstrations are collected per task from the standard egocentric viewpoint via scripted policies. Evaluation involves 100 trials each for the standard egocentric viewpoint and novel egocentric viewpoints sampled within $\Delta x \in [-0.1, 0.1]$m, $\Delta y \in [-0.1, 0.1]$m, $\Delta\theta \in [-10, 10]$ degrees.

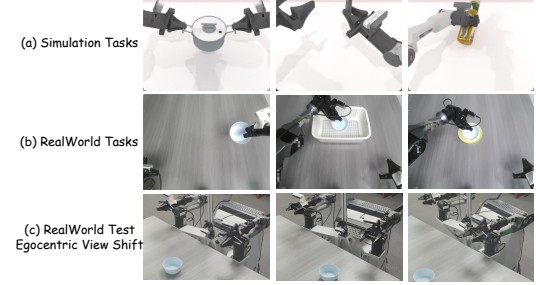

(a) Simulation Tasks

(b) RealWorld Tasks

(c) RealWorld Test Egocentric View Shift

Figure 4: Simulation and Real-World Tasks with Egocentric View Shift.

**Real-World Environment:** We conduct real-world experiments in a Mobile ALOHA platform. Detailed Setup is in Appendix A.7.1. Tasks include *Pick Up Bowl*, *Pick Bowl and Place to Basket*, and *Pick Bowl and Place to Plate*. 50 demonstrations are collected per task from the standard egocentric viewpoint via teleoperation. Evaluation involves 20 trials each for three fixed viewpoints: standard, counterclockwise, and clockwise egocentric viewpoint, as shown in Figure 4(c).

#### 4.1.2 BASELINE METHODS

We evaluate the following baselines under different training-data configurations and use policy model ACT (Zhao et al., 2023) for simulation and $\pi_0$ (Black et al., 2024) for the real robot:

- **Standard View (50):** Training with 50 standard egocentric viewpoint demonstrations
- **Standard View (100):** Training with 100 standard egocentric viewpoint demonstrations (simulation only)
- **EgoDemoGen w/o EgoViewTransfer:** Training with 50 standard + 50 naively composed demonstrations (directly merge scene video and robot video)

#### 4.1.3 IMPLEMENTATION DETAILS

**EgoViewTransfer Models:** We adapt CogVideoX-5b-I2V (Yang et al., 2024) by expanding input channels to 48 for dual-video conditioning. The model is finetuned on a large multi-task demonstration dataset: 380 episodes for simulation (details in Appendix A.5) and 1,000 episodes for real-world (details in Appendix A.7), split 9:1 for training and validation.

For demonstration generation, novel egocentric viewpoints are sampled within $\Delta x \in [-0.1, 0.1]$m, $\Delta y \in [-0.1, 0.1]$m, $\Delta \theta \in [-10, 10]$ degrees relative to the original base pose in simulation and $\Delta x \in [-0.1, 0.0]$m, $\Delta y \in [-0.1, 0.1]$m, $\Delta \theta \in [-10, 10]$ degrees in real-world because real-world robot cannot move forward. Wrist camera observations remain unchanged from the original demonstrations. For 50 standard egocentric demonstrations, we sample one novel egocentric viewpoint per demonstration and **EgoDemoGen** generate novel demonstration.

**Policy Models:** Details of policy models can be found in Appendix A.6 (ACT) and Appendix A.8 ($\pi_0$).

### 4.1.4 EVALUATION METRICS

**Policy Performance:** We evaluate policy performance using Task Success Rate.

**Video Quality:** We evaluate the generated videos using PSNR, SSIM, LPIPS (Zhang et al., 2018), and FVD (Unterthiner et al., 2018) to quantify visual similarity, fidelity, and temporal coherence compared to ground truth videos.

## 4.2 MAIN RESULTS

**Simulation.** Table 1 shows that **EgoDemoGen** achieves the best results on both views. Averaged over tasks, the novel view improves from $14.7\%$ to $30.0\%$ ($+15.3\%$ abs.), while the standard view increases from $78.0\%$ to $80.7\%$ ($+2.7\%$ abs.). Compared with *EgoDemoGen (w/o EgoViewTransfer)*, the gains are $15.3\% \rightarrow 30.0\%$ ($+14.7\%$ abs.) on the novel view and $75.7\% \rightarrow 80.7\%$ ($+5.0\%$ abs.) on the standard view. Per task, novel-view gains are consistent—*Lift Pot* $18\% \rightarrow 43\%$ ($+25\%$ abs.), *Handover Mic* $7\% \rightarrow 22\%$ ($+15\%$ abs.), *Shake Bottle* $19\% \rightarrow 25\%$ ($+6\%$ abs.); the standard view changes are $75\% \rightarrow 83\%$ ($+8\%$ abs.), $94\% \rightarrow 96\%$ ($+2\%$ abs.), and $65\% \rightarrow 63\%$ ($-2\%$ abs.), respectively. Overall, kinematics-based retargeting plus **EgoViewTransfer** yields *paired* demonstrations that enhance viewpoint robustness without degrading average standard-view performance.

Table 1: **Simulation results: success rates (%) across tasks and viewpoints (100 trials each).** Standard: Standard Egocentric View, Novel: Novel Egocentric View.

| Method | Lift Pot | | Handover Mic | | Shake Bottle | | Average | |
|---|---|---|---|---|---|---|---|---|
| | Standard | Novel | Standard | Novel | Standard | Novel | Standard | Novel |
| Standard View (50) | 64 | 16 | 85 | 11 | 42 | 10 | 63.7 | 12.3 |
| Standard View (100) | 75 | 18 | 94 | 7 | **65** | 19 | 78.0 | 14.7 |
| EgoDemoGen (w/o EgoViewTransfer) | 68 | 20 | 94 | 12 | **65** | 14 | 75.7 | 15.3 |
| **EgoDemoGen** | **83** | **43** | **96** | **22** | 63 | **25** | **80.7** | **30.0** |

**Real-world.** Table 2 summarizes three real-world tasks (novel view averaged over CCW/CW; 20 trials per condition). **EgoDemoGen** attains the best averages. Novel-view success rises from $36.7\%$ to $62.5\%$ ($+25.8\%$ abs.) over *Standard View (50)* and from $43.3\%$ to $62.5\%$ ($+19.2\%$ abs.) over *EgoDemoGen (w/o EgoViewTransfer)*. Standard-view success increases from $60.0\%$ to $78.3\%$ ($+18.3\%$ abs.) and from $51.7\%$ to $78.3\%$ ($+26.6\%$ abs.) relative to *w/o EgoViewTransfer*. Gains hold across all three tasks and both rotation directions, indicating that pairing retargeted actions with repaired videos improves egocentric viewpoint robustness without harming standard-view performance.

Table 2: **Real-world results: success rates (%) across tasks and viewpoints (20 trials each).** Std.: Standard Egocentric View, CCW: Counterclockwise View, CW: Clockwise View, Nov.: Novel Egocentric View (averaged over CCW and CW).

| Method | Pick Up Bowl | | | Place Bowl Basket | | | Place Bowl Plate | | | Average | |
|---|---|---|---|---|---|---|---|---|---|---|---|
| | Std. | CCW | CW | Std. | CCW | CW | Std. | CCW | CW | Std. | Nov. |
| Standard View (50) | 70 | 40 | 55 | 65 | 35 | 45 | 45 | 25 | 20 | 60.0 | 36.7 |
| EgoDemoGen (w/o EgoViewTransfer) | 50 | 15 | 40 | 60 | 65 | 75 | 45 | 45 | 20 | 51.7 | 43.3 |
| **EgoDemoGen** | **90** | **75** | **70** | **80** | **75** | **75** | **65** | **45** | **35** | **78.3** | **62.5** |

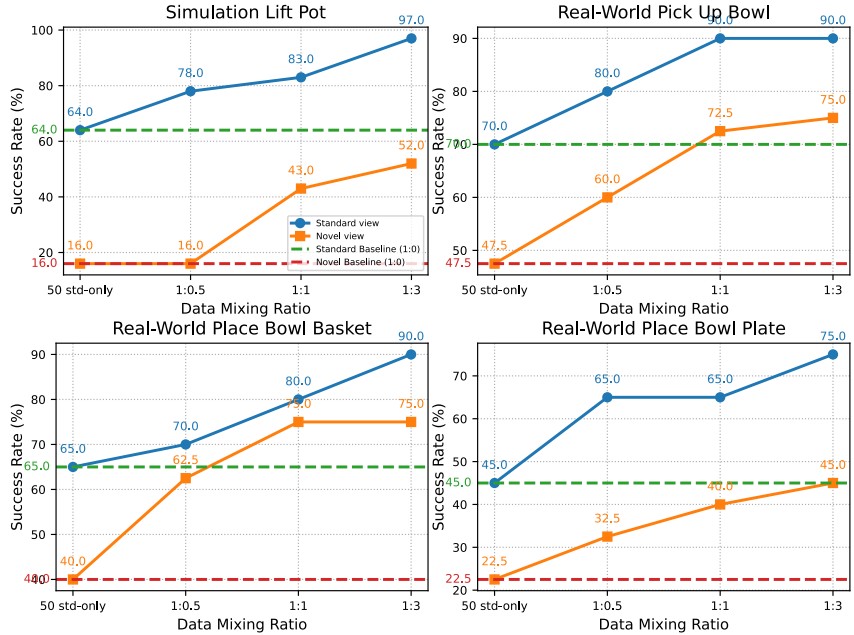

Figure 5: Success Rates under varying data mixture ratios for the standard view and novel egocentric view. The dashed lines indicate the 1:0 baselines, and for real-world results the novel curve is the mean over Counterclockwise/Clockwise rotations.

### 4.3 DATA MIXING RATIO ANALYSIS

Figure 5 plots success under the *standard view* and the *novel egocentric view* across mixing ratios {1:0, 1:0.5, 1:1, 1:3}. Increasing the share of *paired* **EgoDemoGen** demonstrations improves both curves, with larger gains on the novel view on most tasks. A clear knee appears at **1:1**: performance is flat/modest at 1:0.5, jumps at 1:1, then grows more slowly at 1:3 (diminishing returns). Standard-view accuracy rises in parallel—no trade-off—indicating that mixing generated pairs with source data enhances viewpoint robustness without harming in-distribution behavior. Overall, **1:1** is a strong accuracy–cost point, with **1:3** giving smaller additional gains.

### 4.4 ACTION RETARGETING VALIDATION

Table 3 reports replay success in the *novel* egocentric view (mean over 20 random viewpoints). **Retargeted actions** vastly outperform *original actions*, raising the average from 8.3% to 78.3% (+70.0% abs., $\approx$ 9.4$\times$). Per task: *Lift Pot* 5% $\to$ 100%

Table 3: **Novel egocentric-view replay success rate (%).** Evaluated over 20 random views per task.

| Method | Lift Pot | Handover Mic | Shake Bottle | Average |
|---|---|---|---|---|
| Original Action | 5 | 15 | 5 | 8.3 |
| **Retargeted Action** | **100** | **60** | **75** | **78.3** |

(+95% abs.), *Handover Mic* 15% $\to$ 60% (+45% abs.), *Shake Bottle* 5% $\to$ 75% (+70% abs.). These results show that source-frame actions do not transfer to novel viewpoints; kinematics-based retargeting (IK in the novel frame) is essential for feasible trajectories and for constructing paired (video, action) demonstrations that enable viewpoint-robust manipulation.

### 4.5 VIDEO GENERATION QUALITY ANALYSIS

**Quantitative Analysis.** In simulation, we compare synthesized videos against the GT reference under novel egocentric views. As shown in Table 4, *Naive Composition* suffers from artifacts and temporal flicker, while **EgoViewTransfer** achieves markedly lower FVD (460.62$\to$211.99) and LPIPS (0.1175$\to$0.1145), indicating closer alignment with the GT distribution. The slight decrease in PSNR/SSIM is expected in clean simulated environments where pixelwise overlap favors direct merge.

Table 4: **Simulation (Novel Egocentric View): video generation metrics.** Naive Composition: directly merge scene video and robot video without generative repair. Higher ↑ is better; lower ↓ is better.

| Task | Method | PSNR ↑ | SSIM ↑ | LPIPS ↓ | FVD ↓ |
|---|---|---|---|---|---|
| Lift Pot | Naive Composition | **25.998** | **0.9064** | 0.1020 | 387.63 |
| | EgoViewTransfer | 25.916 | 0.8929 | **0.0938** | **157.83** |
| Handover Mic | Naive Composition | 22.404 | **0.9130** | 0.1323 | 617.74 |
| | EgoViewTransfer | **23.077** | 0.9017 | **0.1297** | **272.64** |
| Shake Bottle | Naive Composition | **21.788** | **0.8963** | **0.1182** | 376.48 |
| | EgoViewTransfer | 21.126 | 0.8675 | 0.1201 | **205.49** |
| **Average** | **Naive Composition** | **23.397** | **0.9052** | 0.1175 | 460.62 |
| | **EgoViewTransfer** | 23.373 | 0.8874 | **0.1145** | **211.99** |

In the real world, evaluated on a double reprojection validation set (Table 5), **EgoViewTransfer** improves PSNR and SSIM and substantially reduces LPIPS (0.2453→0.0870) and FVD (896.46→148.61), confirming that generative repair enhances visual fidelity and temporal coherence, which aligns with the observed policy-level gains.

Table 5: **Real World (Validation Set, Standard View): video generation metrics.** Naive Composition: directly merge scene video and robot video without generative repair. Higher ↑ is better; lower ↓ is better.

| Method | PSNR ↑ | SSIM ↑ | LPIPS ↓ | FVD ↓ |
|---|---|---|---|---|
| Naive Composition | 19.9507 | 0.7645 | 0.2453 | 896.46 |
| **EgoViewTransfer** | **26.9332** | **0.8895** | **0.0870** | **148.61** |

**Qualitative analysis.** Figure 6 presents representative frames in simulation and the real world comparing GT, **EgoViewTransfer** and Naive Composition. Naive Composition shows reprojection blur and scene artifacts, and the URDF-rendered robot appears synthetic and inconsistent. In contrast, **EgoViewTransfer** removes blur and artifacts and stylizes the overlaid robot to match the scene, yielding cleaner, action-aligned observation videos.

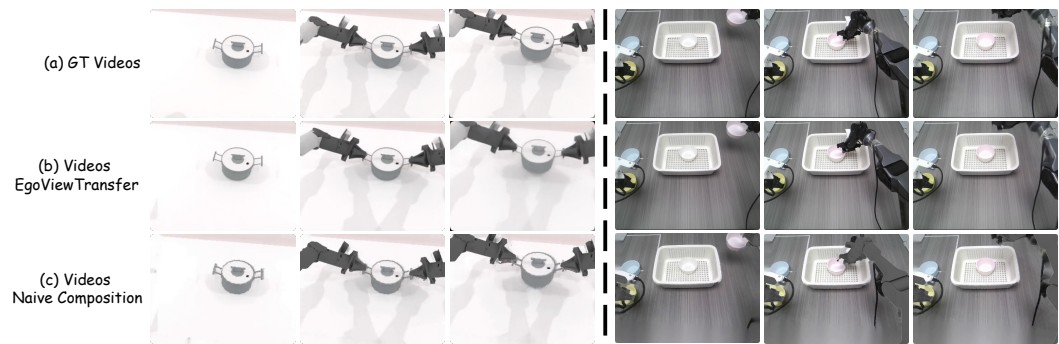

Figure 6: Visualization of observation videos generated by EgoViewTransfer. Left: simulation. Right: real-world.

## 5 CONCLUSION

We present **EgoDemoGen**, a framework that generates *paired* novel egocentric demonstrations by retargeting actions in the novel egocentric frame and synthesizing novel egocentric observation videos with proposed **EgoViewTransfer**, a generative video repair model. Aligning actions and observations in the novel egocentric frame improves success under both novel egocentric viewpoints and source standard egocentric viewpoint. Moreover, performance continues to improve as the proportion of **EgoDemoGen**-generated paired data increases, with diminishing returns. **EgoDemoGen** offers a practical route to improving the egocentric viewpoint robustness of visuomotor policies.

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

# A  APPENDIX

## A.1  THE USE OF LARGE LANGUAGE MODELS (LLMS)

Large language models (LLMs) were used only to aid or polish writing—e.g., grammar correction, wording standardization, and caption/LaTeX cleanup. All edits were manually verified by the authors.

## A.2 KINEMATICS BASED ACTION RETARGETING

**Kinematics-based Action Retargeting** is detailed in Algorithm 1.

---

**Algorithm 1** Kinematics-Based Action Retargeting (FK → Ego Motion → IK)

---

**Input:** joint trajectory $\{\mathbf{q}_t\}_{t=0}^{T-1}$ (14-DoF: L6, Lg, R6, Rg), URDF $\mathcal{U}$, ego-motion $(\Delta x, \Delta y, \Delta \theta)$, IK tolerance schedule $\mathcal{T}$, optional joint mask $\mathcal{M}$
**Output:** retargeted trajectory $\{\tilde{\mathbf{q}}_t\}_{t=0}^{T-1}$

1: Split $\mathbf{q}_t \rightarrow (\mathbf{q}_t^{\mathrm{L}}, \mathbf{g}_t^{\mathrm{L}}, \mathbf{q}_t^{\mathrm{R}}, \mathbf{g}_t^{\mathrm{R}})$
2: **Load robot from $\mathcal{U}$ and instantiate FK/IK solvers**
3: **compute FK:** $T_t^{\mathrm{L}} \leftarrow \mathrm{FK}(q_t^{\mathrm{L}}), \quad T_t^{\mathrm{R}} \leftarrow \mathrm{FK}(q_t^{\mathrm{R}})$
4: $(B_{\mathrm{L}}, B_{\mathrm{R}}) \leftarrow \mathrm{CreateTransform}(\Delta x, \Delta y, \Delta \theta)$
5: **Left targets:** $\hat{T}_t^{\mathrm{L}} \leftarrow B_{\mathrm{L}} \, T_t^{\mathrm{L}}$
6: **Right targets:** $\hat{T}_t^{\mathrm{R}} \leftarrow B_{\mathrm{R}} \, T_t^{\mathrm{R}}$
7: **Robust IK (left):** $(\tilde{Q}^{\mathrm{L}}, s^{\mathrm{L}}) \leftarrow \mathrm{IK\_Trajectory}\Big(\{\hat{T}_t^{\mathrm{L}}\}, \text{init} = q_0^{\mathrm{L}}, \text{schedule} = \mathcal{T}, \text{mask} = \mathcal{M}\Big)$
8: **Robust IK (right):** $(\tilde{Q}^{\mathrm{R}}, s^{\mathrm{R}}) \leftarrow \mathrm{IK\_Trajectory}\Big(\{\hat{T}_t^{\mathrm{R}}\}, \text{init} = q_0^{\mathrm{R}}, \text{schedule} = \mathcal{T}, \text{mask} = \mathcal{M}\Big)$
   ▷ *Trajectory IK uses LM solver with tolerance escalation, random restarts, and interpolate fill for failures.*
9: (Optional) smooth successful joint sequences with median filtering
10: **for** $t = 0$ to $T-1$ **do**
11:    **Compose actions with original grippers:** $\tilde{q}_t \leftarrow [\, \tilde{Q}_t^{\mathrm{L}}, \, g_t^{\mathrm{L}}, \, \tilde{Q}_t^{\mathrm{R}}, \, g_t^{\mathrm{R}} \,]$
12: **end for**
13: **return** $\{\tilde{q}_t\}_{t=0}^{T-1}$

---

## A.3 RENDERING OF ROBOT VIDEO AND MASK

**Rendering of Robot Video and Mask** is detailed in Algorithm 2.

---

**Algorithm 2** Rendering of Robot-Only Video and Mask in the Camera View

---

**Input:** URDF $\mathcal{U}$, joint trajectory $\{\mathbf{q}_t\}_{t=0}^{T-1}$, intrinsics $K$, extrinsic $T_{\mathrm{cam} \leftarrow \mathrm{base}}$, frame rate $f_{\mathrm{fps}}$
**Output:** robot-only video $\mathcal{V}_{\mathrm{robot}}$ (RGB or RGBA) and robot mask $\mathcal{M}_{\mathrm{robot}}$

1: **Load robot from $\mathcal{U}$;** cache link meshes and materials; instantiate FK.
2: **Configure off-screen renderer:** set camera($K, T_{\mathrm{cam} \leftarrow \mathrm{base}}$), image size $(H, W)$, lighting, and z-buffer.
3: **for** $t = 0$ to $T-1$ **do**
4:    **FK:** compute per-link transforms $\{L_i(t)\}$ in the base frame from $\mathbf{q}_t$.
5:    **To camera:** $L_i^{\mathrm{cam}}(t) \leftarrow T_{\mathrm{cam} \leftarrow \mathrm{base}} \, L_i(t)$ for all links $i$.
6:    **Rasterize:** render only robot geometry with z-buffer → $\mathrm{RGB}_t, \mathrm{Mask}_t$ (and optional $\alpha_t$, $\mathrm{Depth}_t$).
7:    **Background:** keep transparent (RGBA) or composite on a constant color.
8: **end for**
9: Stack $\{\mathrm{RGB}_t\}$ (and $\alpha_t$ if present) and encode at $f_{\mathrm{fps}}$ to obtain $\mathcal{V}_{\mathrm{robot}}$.
10: Stack $\{\mathrm{Mask}_t\}$ and encode at $f_{\mathrm{fps}}$ to obtain $\mathcal{M}_{\mathrm{robot}}$.
   ▷ *Off-screen rasterization with anti-aliasing; depth/alpha can be exported for later fusion.*
11: **return** $\mathcal{V}_{\mathrm{robot}}, \mathcal{M}_{\mathrm{robot}}$

---

## A.4 NOVEL VIEWPOINT REPROJECTION OF RGB AND DEPTH

As shown in Algorithm 3, for each frame we align depth to RGB, backproject to camera 0, transform points with $T_{0 \rightarrow 1}$, and z-buffered bilinear-splat them to obtain the novel-view RGB and depth $(\hat{I}_t, \hat{D}_t)$.

---

**Algorithm 3** RGB–D Novel-View Reprojection

---

**Input:** $\{I_t, D_t\}_{t=0}^{T-1}$ (RGB, depth in mm), camera intrinsics $K \in \mathbb{R}^{3\times3}$, extrinsic $T_{0\to1} \in \mathrm{SE}(3)$, optional masks $\{M_t\}$, depth scale $s$ (mm→m), max depth $d_{\max}$

**Output:** $\{\hat{I}_t, \hat{Z}_t\}_{t=0}^{T-1}$: novel-view RGB and depth

1: **for** $t = 0$ to $T - 1$ **do**
2:      $D_t^{\mathrm{rgb}} \leftarrow \mathrm{AlignDepthToRGB}(D_t, I_t, s)$        ▷ align depth to RGB resolution
3:      $(I_t^\star, D_t^\star) \leftarrow \mathrm{ApplyMask}(I_t, D_t^{\mathrm{rgb}}, M_t)$        ▷ optional; with spatial dilation
4:      $(\mathbf{P}_t, \mathbf{C}_t) \leftarrow \mathrm{Backproject}(I_t^\star, D_t^\star, K, s, d_{\max})$        ▷ $\mathbf{P}_t \in \mathbb{R}^{N_t\times3}$, $\mathbf{C}_t \in [0,1]^{N_t\times3}$
5:      $\mathbf{P}_t^{(1)} \leftarrow T_{0\to1}\,\mathbf{P}_t$        ▷ rigid transform to target camera
6:      $(\hat{I}_t, \hat{D}_t) \leftarrow \mathrm{ProjectZBuffer}(\mathbf{P}_t^{(1)}, \mathbf{C}_t, K)$        ▷ bilinear splatting + per-pixel min-depth
7: **end for**
8: **return** $\{\hat{I}_t, \hat{D}_t\}_{t=0}^{T-1}$

---

*Notes.* Depth is assumed in millimeters and converted via $s$. Frames can be processed independently (optionally in parallel across $t$ on CPU for safety), or in a single process that can run on GPU. (Optional) A second backproject–reproject pass via $T_{1\to0}$ can be added if double reprojection is desired.

---

## A.5 SIMULATION IMPLEMENTATION DETAILS

### A.5.1 SIMULATION ROBOT SETUP

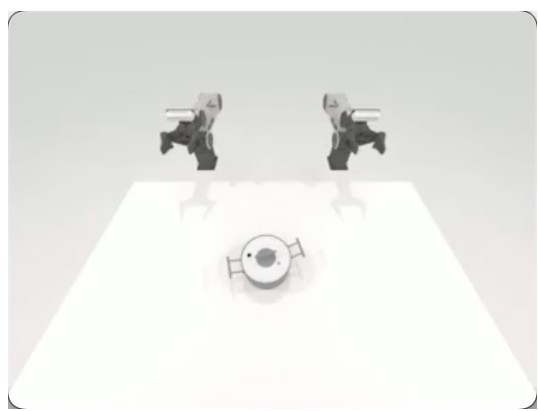

Figure 7: Simulation Robot Setup.

As illustrated in Figure. 7, the RoboTwin 2.0 (Chen et al., 2025b) simulation environment uses two *Piper* manipulators, one head-mounted egocentric camera, and two wrist-mounted cameras (D435; vertical field of view $\mathrm{FOV}_y = 37°$; resolution $240 \times 320$). The distance between the two arms is $0.68\,\mathrm{m}$.

### A.5.2 DETAILS OF SIMULATION EGOVIEWTRANSFER MODEL

We construct the training dataset from 39 simulation tasks: `beat_block_hammer`, `blocks_ranking_rgb`, `blocks_ranking_size`, `click_bell`, `dump_bin_bigbin`, `handover_block`, `handover_mic`, `lift_pot`, `move_can_pot`, `move_pillbottle_pad`, `move_playingcard_away`, `move_stapler_pad`, `open_laptop`, `open_microwave`, `pick_diverse_bottles`, `pick_dual_bottles`, `place_a2b_left`, `place_a2b_right`, `place_bread_basket`, `place_bread_skillet`, `place_burger_fries`, `place_can_basket`, `place_container_plate`, `place_dual_shoes`, `place_empty_cup`, `place_fan`, `place_mouse_pad`, `place_object_basket`, `place_object_scale`, `place_object_stand`, `place_phone_stand`, `place_shoe`, `press_stapler`, `put_bottles_dustbin`, `shake_bottle`, `shake_bottle_horizontally`, `stamp_seal`, and `turn_switch`. Under the *clean* configuration, we collect 10 episodes

per task in the *standard egocentric view*, yielding a total of **390** episodes, which form the training set for the simulation video generative repair model.

For the double-reprojection training data, the sample range of novel egocentric viewpoints is $\Delta x \in [-0.1, 0.1]$ m, $\Delta y \in [-0.1, 0.1]$ m, $\Delta \theta \in [-10, 10]$ degrees.

Model Training uses AdamW optimizer (lr=2e-5), batch size 2/GPU, gradient accumulation over 8 steps, for 120 epochs. Input resolution is 240×320. The number of frames is 49. Model is finetuned on 4×H20 GPUs.

During inference, we process long videos as 49-frame segments, and use 25 denoising steps with DPM scheduler per segment.

### A.6 DETAILS OF ACT

We use default ACT (Zhao et al., 2023) settings unless noted. Batch size is 16, learning rate is $1 \times 10^{-5}$, action chunk size is 50, and input resolution is $480 \times 640$ per camera. Training runs for $40k$ steps on 1×RTX 4090 GPU. During testing, temporal aggregation is used.

### A.7 REAL-WORLD IMPLEMENTATION DETAILS

#### A.7.1 REAL-WORLD ROBOT SETUP

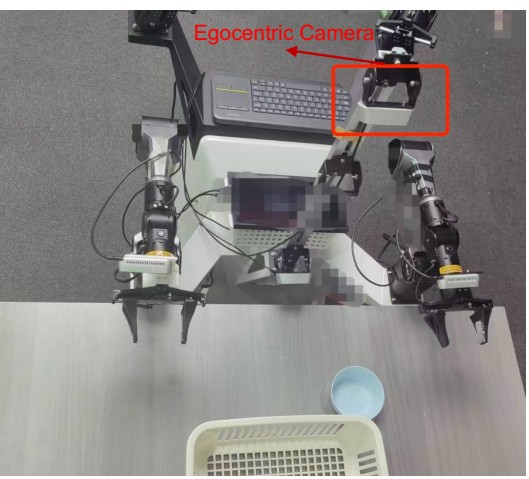

Figure 8: Real-World Robot Setup.

As illustrated in Figure. 8, real-world environment uses Mobile ALOHA platform with dual arms, one head-mounted high egocentric camera, and two wrist-mounted cameras (Intel RealSense D435i RGB-D cameras; resolution $480 \times 640$). The distance between the two arms is $0.6$ m.

#### A.7.2 DETAILS OF REAL-WORLD EGOVIEWTRANSFER MODEL

We construct the training dataset from 13 real-world tasks: `clean_desk_subtasks`, `clean_desk`, `unfold_shirt`, `Toilet_paper`, `pour_drinks`, `put_microwave`, `RG_robot_data`, `Navigation_to_washing_machine`, `throw_bottle_for_sim2real`, `take_cloth`, `put_cloth_basket`, `clean_sink`, and `table_setting_origin`. Each task contains multiple subtask configurations, and we select up to 3 episodes for each subtask, yielding a total of approximately **1k** episodes for training. For episodes where depth information is missing, we generate scale-consistent depth maps using the method of MoGe (Wang et al., 2025b).

For the double-reprojection training data, the sample range of novel egocentric viewpoints is $\Delta x \in [-0.1, 0.1]$ m, $\Delta y \in [-0.1, 0.1]$ m, $\Delta \theta \in [-10, 10]$ degrees.

Training uses AdamW optimizer (lr=2e-5), batch size 1/GPU, gradient accumulation over 8 steps, for 120 epochs. Input resolution is 480×640. The number of frames is 49. Model is finetuned on 4×H20 GPUs.

During inference, we process long videos as 49-frame segments, and use 25 denoising steps with DPM scheduler per segment.

### A.8 DETAILS OF $\pi_0$

We follow default $\pi_0$ (Black et al., 2024) settings unless noted. Batch size is 128, input resolution is $480 \times 640$, and training runs for $40k$ steps on 4×H20 GPUs.

### A.9 MORE VISUALIZATION RESULTS

Visualization of policy execution in the Simulation is shown in Figure 9. Visualization of policy execution in the real world is shown in Figure 10. Visualization of EgoViewTransfer in Simulation is shown in Figure 11. Visualization of EgoViewTransfer in Real-World is shown in Figure 12.

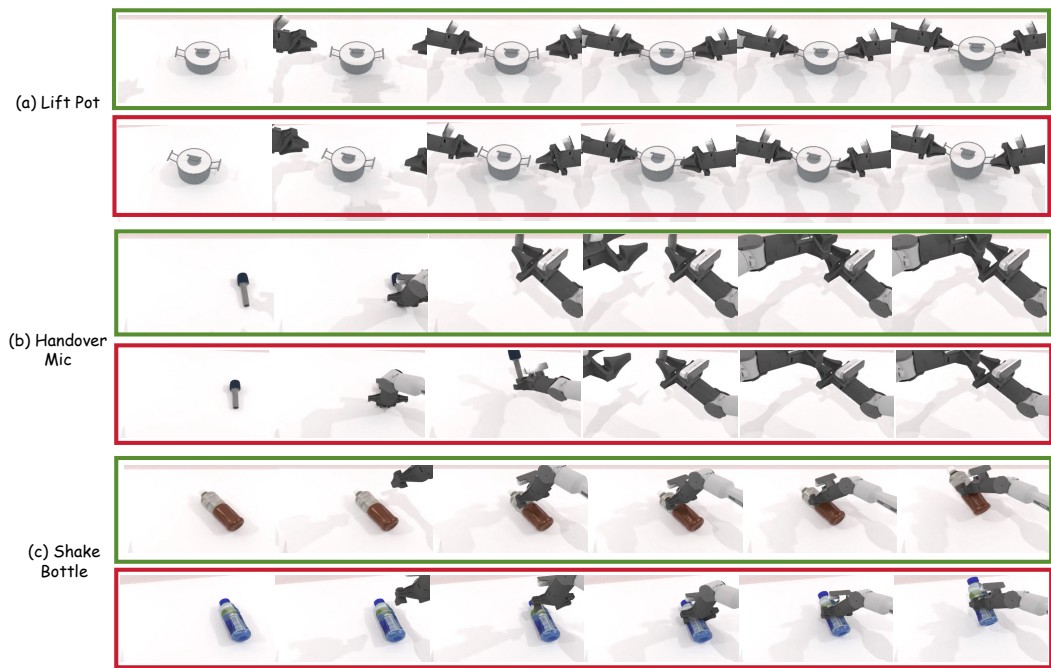

Figure 9: Visualization of policy execution in the Simulation. The **green** boxes denote the standard egocentric view, the **red** boxes denote the random novel egocentric view.

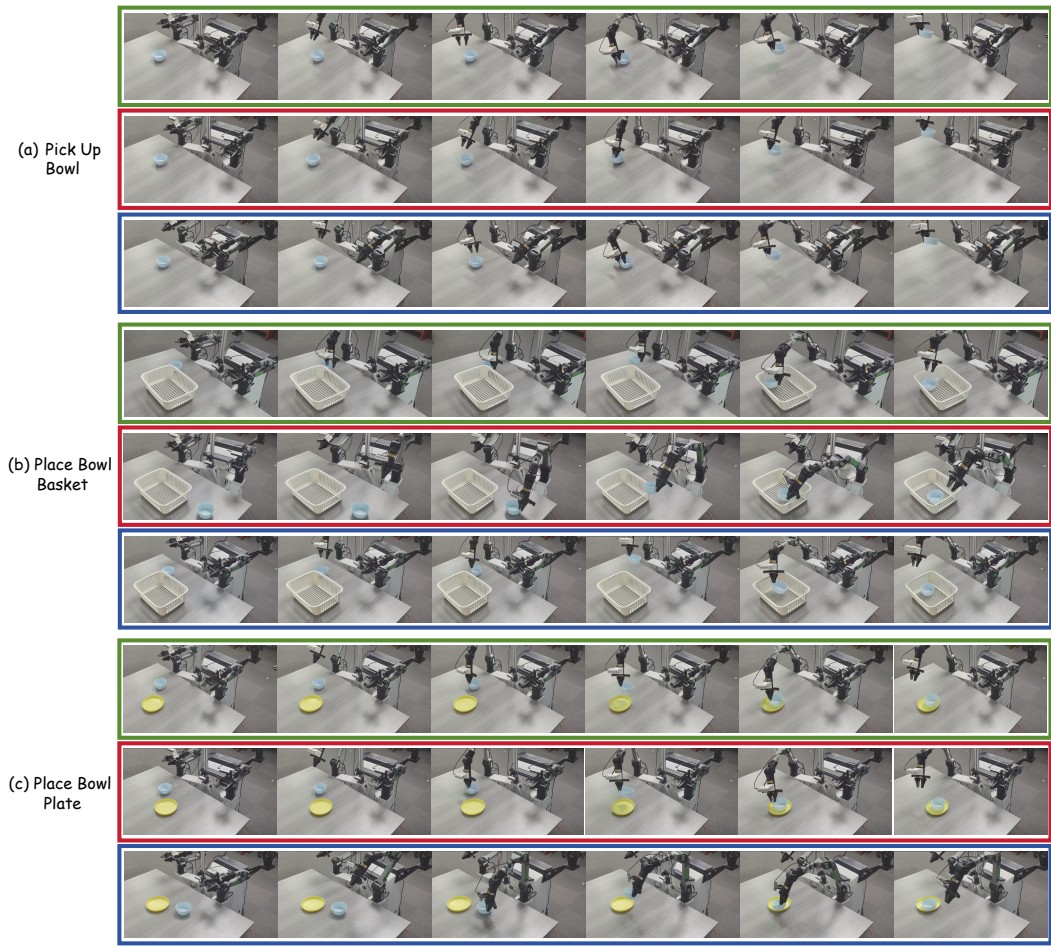

Figure 10: Visualization of policy execution in the real world. The **green** boxes denote the standard egocentric view, the **red** boxes denote the counterclockwise egocentric view, and the **blue** boxes denote the clockwise egocentric view.

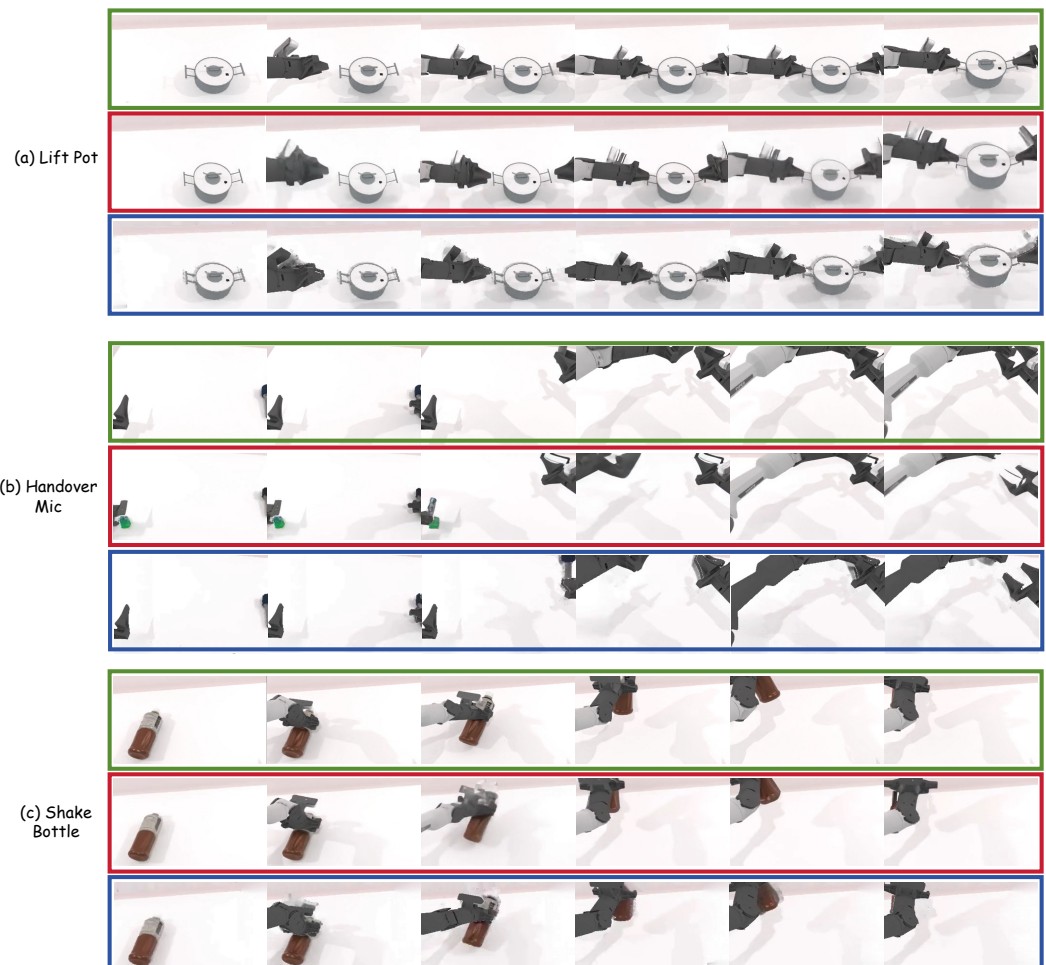

(a) Lift Pot

(b) Handover Mic

(c) Shake Bottle

Figure 11: Visualization of EgoViewTransfer in Simulation. The **green** boxes denote the GT video, the **red** boxes denote the Video w/ EgoViewTransfer, and the **blue** boxes denote the Video w/o EgoViewTransfer (Naive Composition).

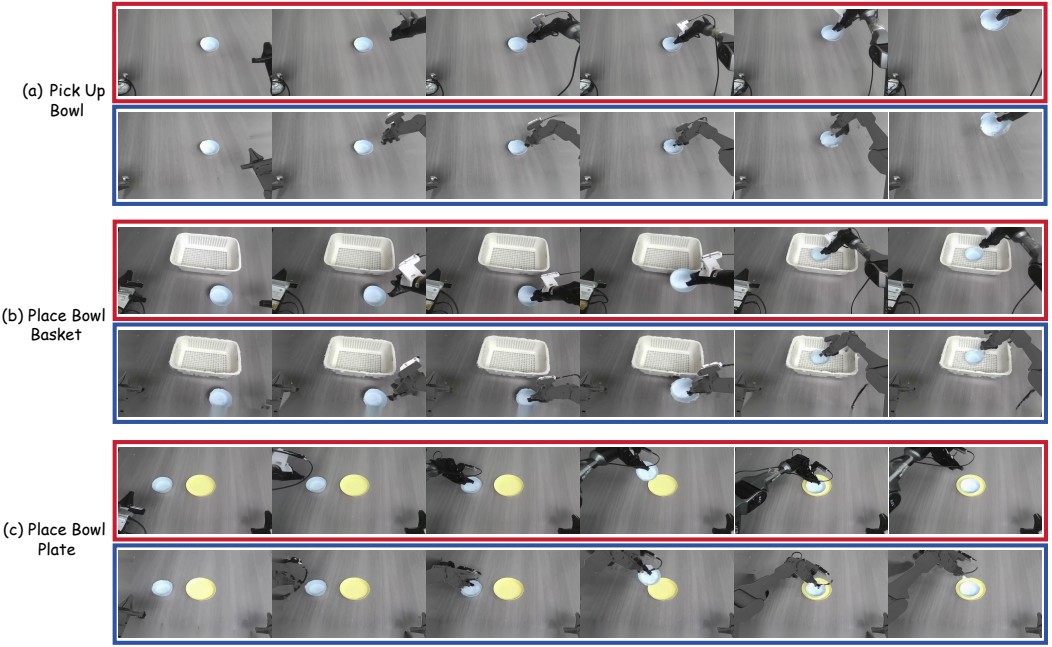

Figure 12: Visualization of EgoViewTransfer in Real-World. The **red** boxes denote the Video w/ EgoViewTransfer, the **blue** boxes denote the Video w/o EgoViewTransfer (Naive Composition).

