# OpenReview forum: "EgoDemoGen: Novel Egocentric Demonstration Generation Enables Viewpoint-Robust Manipulation"
_ICLR.cc/2026/Conference — ICLR 2026 Conference Withdrawn Submission_

### Official Review · Reviewer_mtCA · 2025-10-28

**Soundness:** 3
**Presentation:** 3
**Contribution:** 2
**Rating:** 4
**Confidence:** 3

**Summary:**

The paper presents EgoDemoGen, a framework that synthesizes demonstrations from novel egocentric viewpoints to improve policy generalization. To collect a kinematically feasible action sequence, EgoDemoGen first retargets the actions from the original demonstration in the novel egocentric frame via kinematic transformation. Then, to generate an egocentric video that matches the retargeted actions, the authors choose to separately construct a novel viewpoint scene video as well as a robot-only video, then recombine the two videos through a video repair model, EgoViewTransfer. The repair model is finetuned to reconstruct the original video given the double-projected scene video and the source robot video. Experiments showcase that the generated demonstrations greatly improve the policy's robustness to novel viewpoints, in both simulated and real environments.

**Strengths:**

- The authors propose a framework that generates egocentric demonstrations for novel viewpoints based on demonstrations collected from a single viewpoint, providing an effective solution to augment robot demonstrations with various egocentric viewpoints.
- Experimental results highlight that the generated demonstrations improve the policy performance under both standard and novel viewpoints, and across simulated and real-robot setups
- The paper is well organized and easy to read.

**Weaknesses:**

- Most novel viewpoints (e.g. $\Delta$x ∈ [−0.1, 0.1]) appear to be small perturbations on the standard viewpoint. Although visuomotor policies can already be sensitive to small observation shifts, I wonder if the framework can remain robust when the novel viewpoints have a larger distinction from the original one.
- To demonstrate the effectiveness of double reprojection strategy, one ablation study should compare double-reprojected scene videos against source scene videos when used to finetune the video repair model.
- Do the viewpoints sampled by EgoDemoGen overlap with the viewpoints used for evaluation? If so, it would be helpful to investigate whether the policy will naturally have stronger generalizability to truly unseen viewpoints after training on demonstrations from more viewpoints.

**Questions:**

- Typo: “vide -> "video” in line 087, “repeoject" -> "reproject” and “he” -> “the” in line 265
- Why does the performance of the standard viewpoint benefit more from demonstrations of novel viewpoints than those of the standard viewpoint in the simulated setup (e.g., EgoDemoGen vs. Standard View (100) )?
- It would be very helpful if the authors could elaborate more on the experimental setups for Table 3.
- What is Naive Composition? How are the scene and robot videos merged directly?
- What is the average length/number of frames of robot videos?

---

### Official Review · Reviewer_ozfS · 2025-10-31

**Soundness:** 2
**Presentation:** 3
**Contribution:** 2
**Rating:** 2
**Confidence:** 3

**Summary:**

The paper addresses the problem of learning manipulation policies from egocentric viewpoints. To improve robustness against viewpoint shifts, the authors propose a model that synthesizes observations from novel perspectives along with the corresponding robot actions.

**Strengths:**

The paper tackles an important aspect of policy generalization — robustness to changes in observation viewpoints.

The proposed framework is evaluated with both simulation and real-world hardware.

**Weaknesses:**

Given the presence of several related works that employ retargeting for generating robot actions and inpainting for synthesizing novel observations [1, 2, 3], the novelty and technical contribution of this work appear limited.

The proposed EgoViewTransfer model requires in-context robot data for fine-tuning, which constrains its applicability to new tasks and unseen scenarios.

The comparison with existing baselines appears insufficient; additional relevant works should be included for a more comprehensive evaluation and discussion.


[1] Masquerade: Learning from In-the-wild Human Videos using Data-Editing, arXiv’25;

[2] Phantom: Training Robots Without Robots Using Only Human Videos, CoRL’25;

[3] MotionTrans: Human VR Data Enable Motion-Level Learning for Robotic Manipulation Policies, arXiv’25.

**Questions:**

Could perturbations in the robot base pose lead to infeasible inverse kinematics (IK) solutions? If so, how do the authors handle such cases?

Have the authors explored zero-shot deployment --- that is, training the policy solely on synthetic data and directly deploying it in the real world without using real robot data?

---

### Official Review · Reviewer_sf96 · 2025-11-01

**Soundness:** 3
**Presentation:** 3
**Contribution:** 2
**Rating:** 4
**Confidence:** 3

**Summary:**

This paper proposes the EgoDemoGen algorithm, a new framework that generates paired egocentric demonstrations from novel viewpoints in robotic manipulation. The method combines kinematics-based action retargeting with a generative video repair model, enabling viewpoint-consistent observation–action pairs. The approach is evaluated in both simulation (RoboTwin2.0) and real-world settings, showing consistent improvements in policy success rates when training with the generated data.

**Strengths:**

1. The paper shows an important gap in imitation learning in robotic manipulation, that is the sensitivity to egocentric viewpoint changes, and motivates the need for viewpoint-robust data generation.
2. The authors evaluate on both simulation and real-world environments, with multiple tasks and detailed quantitative comparisons. The improvements in policy success rate are significant and consistently reported.

**Weaknesses:**

1. The paper’s technical contribution is incremental relative to recent advances of Demogen [1], Rovi-Aug [2], and Gaussian-splatting-based demonstration synthesis [3]. EgoDemoGen mainly combines these known results without introducing a distinct algorithmic innovation or theoretical insight. The “video repair” formulation is vaguely defined and not clearly differentiated from existing conditional video diffusion approaches.
2. The baselines are minimal and do not include any strong or recent viewpoint-augmentation methods, making it unclear how the proposed approach compares to the state of the art.
3. The paper reports quantitative improvements but offers little qualitative or mechanistic explanation of why EgoDemoGen works. There is no analysis of failure cases, sensitivity to viewpoint displacement, or robustness under more extreme camera shifts. The claimed “diminishing returns” trend is mentioned but not analyzed in depth, e.g., no discussion of data quality degradation, overfitting, or diversity issues.

[1] Zhengrong Xue, Shuying Deng, Zhenyang Chen, Yixuan Wang, Zhecheng Yuan, and Huazhe Xu.
Demogen: Synthetic demonstration generation for data-efficient visuomotor policy learning. arXiv
preprint arXiv:2502.16932, 2025.
[2] Lawrence Yunliang Chen, Chenfeng Xu, Karthik Dharmarajan, Richard Cheng, Kurt Keutzer,
Masayoshi Tomizuka, Quan Vuong, and Ken Goldberg. Rovi-aug: Robot and viewpoint augmentation
for cross-embodiment robot learning. In Conference on Robot Learning, pp. 209–233. PMLR,
2025a.
[3] Sizhe Yang, Wenye Yu, Jia Zeng, Jun Lv, Kerui Ren, Cewu Lu, Dahua Lin, and Jiangmiao Pang.
Novel demonstration generation with gaussian splatting enables robust one-shot manipulation.
arXiv preprint arXiv:2504.13175, 2025.

**Questions:**

1. How sensitive is the proposed method’s performance to the specific pretrained backbone, e.g., CogVideoX-5B?
2. Can the system generalize to viewpoint transformations larger than ±10° or 10 cm translations?
3. Would training the policy entirely on generated data (no real egocentric demos) still yield usable performance?

---

### Official Review · Reviewer_St7q · 2025-11-01

**Soundness:** 2
**Presentation:** 2
**Contribution:** 3
**Rating:** 4
**Confidence:** 3

**Summary:**

This paper aims at multi-viewpoint demonstration generation in the egocentric robot manipulation scenario, to the end of more robust policy generalization under viewpoint shift. This work proposes a two-step strategy for generating new egocentric manipulation demonstrations that preserve the robot-camera-pose relativity: (a). First, a new action sequence that is deployed at the new viewpoint to reach the same goal is inferred (Action Retargeting); (b). Second, a new sequence of observations (which fuses two standalone videos: a scene video and a robot-only video) at the new viewpoint is generated (EgoViewTransfer).

**Strengths:**

1. The paper incorporates robotics (kinematics) and computer vision techniques (segmentation video rendering, double reprojection, etc.) to imbue physical configuration conservation during viewpoint change, which is not commonly seen in current video generation works. Essentially, taking kinematics into account enhances physics awareness of the video generation.

2. The experiments are relatively thorough. Sections 4.4 and 4.5 serve as a good study of the proposed modules other than the top-level success rate improvement, in terms of video quality.

3. This framework is self-supervised, meaning no extra data is required given a new viewpoint. In this sense, the method has better potential to scale, compared to those that requires new ground truth data.

**Weaknesses:**

1. The writing of the introduction can be largely improved. The outline in the introduction is not very easy to follow. The research gap (research question) and the novelty in this work should be polished and highlighted in a clearer way.

2. The experiment **setup** is relatively simple, in that the type of tasks is scarce. Is it possible to test the method with more tasks? Additionally, the viewpoint shift is generally small, with 0.2 m in x-y translations and 20 deg in rotation. Under this setting, the difference with even the ground truth videos can be delicate. How would the framework work with larger deviations? Would the performance improvement preserve?

3. The performance of the proposed approach  was not compared with other video generation baselines.

**Questions:**

* **TECHNICAL QUESTIONS:**

1. Lines 324-326: The x-axis translation for real-world robot is \[-0.1,0] m  because it cannot move forward. In such case, is it fairer to compare with x-axis translation of \[-0.2,0.0] m? Because that smaller variance brings considerably more similarity in the videos. Moreover, in Appendix 7.2, why is it said that the sample range of novel egocentric viewpoints is $\\Delta x \\in \[-0.1,0.1]$?

2. I skimmed through the manuscript (along with the appendix), but cannot find technical details regarding hole filling. Please describe that briefly in the text. Specifically, do you make the assumption that the robot arms do not occlude important information? If important information such as artifact pose is not observable, how can it be inferred during hole filling?

3. For the data mixing ratio analysis (Section 4.3), what are the absolute values of demonstrations in the case of each ratio? Do you fix the total number of demos or fix the number of standard viewpoint demos? How can it be told if the improvement is indeed brought by the change of data mixing ratio, or simply by the increased number of demonstrations?

4. Please show results with other multi-viewpoint video generation methods with the experiment setup in this work, so that we can better compare the performance.

* **PRESENTATION:**

1. Line 78: Is this sentence (led be the colon) a statement of the key insight or the fundamental gap? If this is the gap, the sentence should be rephrased; If is the key insight, then what is the gap? Currently, the writing is confusing.

2. In section 3.1, it is better to mention that without losing generalisability, this work focuses on 2D transformation of the viewpoint (i.e., 3 DoF but not 6 DoF).

3. One general remark is, it would be nicer to have a formality description of the trans-viewpoint demo generation objective in section 3.1. For example, 'the objective is to learn a retargeted action $\\widetilde{Q} \\sim \\pi\_{v'}(a \\mid goal)$, given $\\pi\_{v}(a \\mid goal) \\ s.t., Q \\sim \\pi\_{v}(a \\mid goal)$ (step 1), and then learn or finetune $p(V,Q \\mid \\pi\_{v'})$ (roughly step 2), where $v$ and $v'$ denote different viewpoints'.

4. Line 87, typo: 'generative *video* pair model', not 'vide'.

5. Line 103, grammar: 'consistent, realistic ...' should be 'consistent *and* realistic ...'

6. Line 149, typo: '*an*' egocentric.

7. In equation 2, the homogeneous transformation matrix denoting the transformation from source frame to target frame should be $T\_a^v$, but not $T\_v^a$ (formality convention issue).

8. Line 265, typo: '...*reproject* this novel view ... *the* original source view...'

9. In Figures 1, 2, and 3, the difference in several plots are nuanced. Their difference should be highlighted in some way that is clearer to the readers.

10. Line 276, is $\\epsilon$ a Gaussian noise? If so, it should be mentioned explicitly. Additionally, how are $\\alpha\_{\\tau}$ and $\\sigma\_{\\tau}$ defined?

11. In Equation 6, a rigid bracket is missing.

12. Line 449, a comma before the second 'and' would be good grammar-wise.

---

### Note · Authors · 2025-11-14

I have read and agree with the venue's withdrawal policy on behalf of myself and my co-authors.